# Modelling the Cost-Effectiveness of Implementing a Dietary Intervention in Renal Transplant Recipients

**DOI:** 10.3390/nu13041175

**Published:** 2021-04-02

**Authors:** Friso B. Coerts, Judith J. Gout-Zwart, Eke G. Gruppen, Yvonne van der Veen, Maarten J. Postma, Stephan J. L. Bakker

**Affiliations:** 1Department of Health Sciences, University Medical Center Groningen (UMCG), University of Groningen, Hanzeplein 1, 9713 GZ Groningen, The Netherlands; frisocoerts@gmail.com (F.B.C.); m.j.postma@rug.nl (M.J.P.); 2Department of Nephrology, University Medical Center Groningen (UMCG), University of Groningen, Hanzeplein 1, 9713 GZ Groningen, The Netherlands; e.g.gruppen@umcg.nl (E.G.G.); y.van.der.veen@umcg.nl (Y.v.d.V.); s.j.l.bakker@umcg.nl (S.J.L.B.); 3Asc Academics, Professor Enno Dirk Wiersmastraat 5, 9713 GH Groningen, The Netherlands; 4Department of Economics, Econometrics and Finance, University of Groningen, Nettelbosje 2, 9747 AE Groningen, The Netherlands

**Keywords:** renal transplant recipients, kidney failure, dietary approaches to stop hypertension, diet, cost analysis, cost effectiveness, food technology

## Abstract

Background: The Dietary Approach to Stop Hypertension (DASH) and potassium supplementation have been shown to reduce the risk of death with a functioning graft (DWFG) and renal graft failure in renal transplant recipients (RTR). Unfortunately, a key problem for patients is the adherence to these diets. The aim of this study is to evaluate the cost-effectiveness and budget impact of higher adherence to either the DASH or potassium supplementation. Methods: A Markov model was used to simulate the life course of 1000 RTR in the Netherlands. A societal perspective with a lifetime time horizon was used. The potential effect of improvement of dietary adherence was modelled in different scenarios. The primary outcomes are the incremental cost-effectiveness ratio (ICER) and the budget impact. Results: In the base case, improved adherence to the DASH diet saved 27,934,786 and gained 1880 quality-adjusted life years (QALYs). Improved adherence to potassium supplementation saved €1,217,803 and gained 2901 QALYs. Both resulted in dominant ICERs. The budget impact over a five-year period for the entire Dutch RTR population was €8,144,693. Conclusion: Improving dietary adherence in RTR is likely to be cost-saving and highly likely to be cost-effective compared to the current standard of care in the Netherlands.

## 1. Introduction

Chronic kidney disease (CKD) is a disease characterised by a gradual loss in the functioning of the kidneys, occurring over a period of months or years. The worldwide prevalence of CKD is estimated to be 13.4% [1]. In 2015, CKD led to 1.2 million deaths, which is an increase of 31.7% compared to 2005 [2]. If the kidneys reach an estimated glomerular filtration rate (eGFR) <15 mL/min, they fail to reach a rate of filtration high enough for the patient to survive. This state is labelled end-stage renal disease (ESRD).

For patients with ESRD, treatment is limited to renal replacement therapy (RRT) by either dialysis or kidney transplantation. Dialysis is an artificial method of replacing kidney function. The two most commonly used types are haemodialysis (HD) and peritoneal dialysis (PD). Both types of dialysis remove excess water and waste from the blood: one externally, one internally. Both types of dialysis come with a high costs to society and a low quality of life [3]. Dialysis is estimated to cost between 80,000 and 110,000 euros annually, depending on the type of dialysis [4]. A kidney transplantation has been shown to be cheaper, while patients survive longer and their quality of life is higher [5]. The costs for the procedure of kidney transplantation and the first year of care are estimated to be between 60,000 and 80,000 euros, followed by annual maintenance costs of approximately 10,000 euros [4,6,7]. Unfortunately, there are only limited numbers of kidneys available for transplantation and there is a gap between supply and demand [8,9].

Once a transplanted kidney fails, a patient will require dialysis or re-transplantation to survive. Since the annual costs for dialysis are higher and the quality of life is lower, every extra day that the transplanted kidney stays functional is beneficial. Prevention of need for re-transplantation by improving graft survival will translate into relief of existing organ shortage. Recent observational studies suggest that a patient’s dietary habits affect the rate of mortality and rate of renal graft failure [10,11]. Adhering to the Dietary Approach to Stop Hypertension (DASH) diet has been shown to reduce the risk of death with a functioning graft (DWFG) and renal graft failure [12]. The DASH is a diet consisting of a low intake of red processed meats, sodium, and sweetened beverages and a high intake of legumes, nuts, low-fat dairy products, fruits, vegetables, and whole grains [13]. Similar effects have been found for a higher intake of potassium [14].

The DASH diet and potassium supplementation can be directly adapted into a dietary recommendation for patients. Unfortunately, a key problem for patients is adhering to a diet [15,16]. In current practice, patients usually receive dietary advice once, and, without frequent visits to a dietician, the methods to track the food intake of patients are often self-reported dietary surveys, which can be biased and possibly misleading [17].

We hypothesise that renal transplant recipients (RTR) who adhere to either the DASH diet or potassium supplementation will extend the time until both DWFG and renal graft failure, as well as potentially save costs. The aim of this study is to evaluate the cost-effectiveness and budget impact of higher adherence to either the DASH or potassium supplementation.

## 2. Materials and Methods

An economic evaluation was performed to evaluate the costs and benefits of higher adherence to the DASH diet and potassium supplementation for RTR in the University Medical Center Groningen (UMCG), the Netherlands. The cohort used for this analysis was based on the RTR population of the UMCG (Table 1), a part of the TransplantLines cohort (NCT03272841) [18]. While the actual cohort size was 632 subjects, economic evaluation was performed for 1000 RTR.

This economic evaluation consisted of a cost-effectiveness and budget impact analysis (BIA). The focus of the analyses was the implementation of the DASH diet and potassium supplementation to reduce DWFG and renal graft failure. To address dietary adherence, it was assumed that the number of visits to a dietician was increased from one to eight. The amount of dietician visits in this model is arbitrary as there are no data available regarding the effect on adherence of increased visits to a dietician. It is also possible to further increase the number of visits or for example use mobile applications. Normal adherence to the diet intervention was compared with increased adherence. The cost differences and clinical differences were assessed, and if not dominant (lower costs and clinically superior), the ratio of the differences was presented as the incremental cost-effectiveness ratio (ICER) cost per quality-adjusted life year (QALY) gained. The societal perspective was used in order to address all relevant costs and effects.

### 2.1. The Model

A cohort Markov model was developed using Microsoft Excel 365 (Microsoft Corporation, Redmond, WA, USA). This Markov model was used to simulate the life course of RTR, represented in a Markov trace. As recommended by the Dutch guidelines for economic evaluations in healthcare [19], a time horizon was chosen covering the expected lifetime with a cycle length of one year. Costs were discounted by 4% and health outcomes by 1.5% [19]. The model consists of three main health states through which the RTR transition (Figure 1):(1)Functioning graft(2)Graft failure(3)Death

Patients were able to receive a re-transplantation, and, to include this in the model, extra states were added to differentiate between the first time they are in a state (FG1, GF1) and the second time (FG2, GF2) (Figure 1). It was only possible to receive one re-transplantation given that receiving a third kidney is very unlikely [20]. Notably, in the TransplantLines cohort study, patients with a third kidney only made up only 0.6% of transplanted patients and similar numbers are mentioned in literature [21]. The relatively high age of the patient cohort also reduces their chances of receiving a third kidney. Like the initial transplant, each re-transplant has a probability of primary non-function, which results in transition from GF1 to GF2. To address the fact that events tend to take place in the middle of the year, a mid-cycle correction was applied. As the study population was the RTR population of the UMCG, all patients started in the functioning graft state (FG1).

### 2.2. The Intervention

For the economic evaluation, the DASH diet and potassium supplementation were tested as separate interventions. The dietary intervention consisted of increasing the number of visits to the dietician from one to eight dietician visits. Patients in the RTR population of the UMCG were placed in one of three groups according to their adherence to the DASH diet, similarly they were placed in one of three groups according to their intake of potassium [12]. The potassium intake was based on a 24 h urine sample. Dietary intake for calculating adherence to the DASH diet was determined using a validated semi-quantitative food frequency questionnaire [22]. Adherence to the DASH diet was subsequently calculated using a previously described method which scores the DASH diet adherence ranging from 8 to 40 [13]. The cut-off values for each group were determined by splitting the RTR population for each diet into three equally sized groups. For the DASH diet it was impossible to create three equally sized groups due to an uneven distribution. For both interventions, group 1 had the lowest adherence/intake, group 2 reflected the average, and group 3 had the highest. The mean daily potassium intake was 1873 mg, 2772 mg, and 3882 mg for group 1, group 2, and group 3, respectively. The mean adherence to the DASH diet was scored at 18.5, 24.0, and 29.5 for group 1, group 2, and group 3, respectively. Patients in group 1 had the highest risk of DWFG and renal graft failure compared to group 2 and 3. For all groups of both diets, a separate Markov trace was calculated where only their respective transition probabilities for DWFG and renal graft failure differ. This resulted in a total of six Markov traces, each with a total cost and total QALYs: three traces for the DASH diet and three for the potassium supplementation diet.

It was assumed that only the patients in group 1 would receive the dietary intervention, as it might be expected that greatest benefit could be achieved for these patients. Therefore, in the situation without any dietary intervention 100% of patients in the model are in group 1. With the implementation of the dietary intervention, more patients would follow the DASH diet or potassium supplementation, which was assumed to result in fewer patients in group 1 and more patients in group 2 and 3. The patients from group 1 were equally divided across group 2 and group 3. In the model it is assumed that this transition to another group happens instantly, assuming an immediate effect of the dietary intervention. For the cost-effectiveness analysis, the percentage of patients in group 1 was varied from 0% to 100%. Table 2 provides an overview of the adherence of the three groups to the DASH diet, and for potassium supplementation. For both diets, incremental costs and effects were determined separately.

Incremental costs or savings for RTRs on the DASH diet were determined by calculating the total costs for both the situation without intervention and with intervention using Equation (1), the percentages in Table 3, and the three Markov traces for the DASH diet. Incremental QALYs for the DASH diet were calculated similarly using Equation (2). To calculate incremental costs and QALYs for RTRs on potassium supplementation, Equations (1) and (2) were used, together with Table 2 for the percentages and the three Markov traces for potassium supplementation.
(1)$$\begin{array}{ll} Total\;costs\;=\; & \left(\%\;Group\;1\;\times \;total\;costs\;Group\;1\right)\;+\;\left(\%\;Group\;2\;\times \;total\;costs\;Group\;2\right)\;+\;\\ & \left(\%\;Group\;3\;\times \;total\;costs\;Group\;3\right) \end{array}$$
(2)$$\begin{array}{ll} Total\;QALYs\;=\; & \left(\%\;Group\;1\;\times \;total\;QALYs\;Group\;1\right)+\left(\%\;Group\;2\;\times \;total\;QALYs\;Group\;2\right)\;+\;\\ & \left(\%\;Group\;3\;\times \;total\;QALYs\;Group\;3\right) \end{array}$$

### 2.3. Transition Probabilities

The probabilities at which patients moved from one state to the other were determined using patient data from the UMCG cohort and literature (Table 3). The probability of DWFG and renal graft failure were based on the Kaplan Meier survival curves (using time to graft failure and DWFG data), with survival analysis performed using R and RStudio [25]. These survival curves were fitted with a Weibull distribution, based on the best visual fit and Akaike information criterion (AIC) and Bayesian information criterion (BIC) scores, and the parameters of the distribution were used to extrapolate the survival curves to the length of the model (Appendix A). This analysis was performed separately for each of the three groups for both the potassium supplementation and the DASH diet. From these survival curves the annual transition probabilities were calculated. When the extrapolation crosses the background mortality of the Dutch population, it was assumed that the probabilities would converge; this method was used as the baseline for the model [26].

The probability of re-transplantation and death on dialysis were based on literature [23,24]. For the re-transplantation probability, it was assumed that the probability would linearly decline after age 65 to zero at age 80.

### 2.4. Costs

The perspective of this analysis was a societal perspective; therefore, both direct medical costs and indirect costs were accounted for (Table 4). All costs were adjusted for inflation to reflect 2017 euros; this index year was chosen because the main dataset used for this study was updated to 2017 [18]. Where relevant, costs were acquired from the Dutch costing manual for health economic evaluations [27]. The health states functioning graft and graft failure had corresponding costs for transplantation upkeep and dialysis, respectively. Transitioning from one health state to another also has associated costs: cost of graft failure, transplantation, and death. The friction cost method was used to assess productivity losses, as recommended by the Dutch guidelines. The percentage of patients working in the functioning graft state was used as a reference. When a patient moves from the functioning graft state to the graft failure state, this would incur costs due to productivity loss, but when a patient moves from the graft failure state to the functioning graft state, this would result in savings due to productivity gain. Dietary costs associated with increased adherence to group 2 and group 3 were also included. Dutch data regarding the extra cost of the DASH diet were unavailable, however an article from the United Kingdom estimated the extra costs at around 18% comparing the lowest quintile to the highest [20]. Based on the expert opinion of the authors, the extra costs associated with the DASH diet were estimated at one euro per day. When comparing this to the average that Dutch citizens spend on food (six to seven euros per day), this amounts to an increase between approximately 14 and 17% [28]. The extra costs for potassium supplementation were calculated based on the potassium deficit that group 1 (mean daily intake of 1873 mg) had compared to group 2 and group 3 (a daily deficit between 899 mg and 2009 mg). As the Dutch daily recommended intake of potassium is 3500 mg and the Institute of Medicine recommends an even higher intake of 4700 mg per day, this can be considered a deficit [29,30]. Dietary costs are only incurred while a patient has a functioning graft, since these dietary changes should be avoided while on dialysis.

Increased potassium intake requires monitoring of the potassium levels, and it was assumed that these costs are included in the transplantation upkeep costs, as these include laboratory and diagnostic costs [23]. Since monitoring potassium levels is already part of standard procedure, we assumed that it would not be necessary to use separate costs for monitoring the potassium levels.

For each dietary group (Markov trace), all costs were summed up annually and then discounted. This led to a total cost for each of the groups. Considering the ratio of the groups (i.e., the adherence and the number of patients in each group), the costs were summed up accordingly.

### 2.5. Utilities

Utility values for each of the health states were used to determine the clinical effectiveness of the new intervention, expressed in QALYs (Table 5). For the graft failure state, the health utility value was calculated with regard to the proportion of patients on HD (86%) and PD (14%) [35]. The utility values were calculated using the EQ-5D score. For each dietary group (Markov trace), the utility values were summed up annually and then discounted, resulting in the total amount of QALYs for each group. Considering the ratio of the groups (i.e., the adherence and the number of patients in each group), the QALYs were summed up accordingly.

### 2.6. Sensitivity Analyses

To analyse the robustness of the model and the results, a multivariate probabilistic sensitivity analysis (PSA) and a univariate deterministic sensitivity analysis (DSA) were performed. An adherence for group 1 of 0% was assumed for both the DASH diet and potassium supplementation. The DSA consisted of varying each parameter separately between the lowest and highest value of the 95% confidence interval. If the confidence interval was not available, a standard error of 25% of the parameter’s value was assumed, which was then used to calculate the 95% confidence interval. For the sensitivity analysis of the Weibull extrapolation, a Cholesky decomposition was calculated, from which the variance was determined. Results of the DSA analysis were plotted in a tornado diagram. For the PSA, all parameters were varied based on their respective distributions: beta distributions for the utilities, and percentage working; a gamma distribution for all the costs; and a normal distribution for the time spent in hospital, number of visits to the dietician, and transition probabilities. For the three dietary groups, Monte Carlo simulations were performed with 10,000 iterations each. Results were plotted in a cost-effectiveness plane with a willingness-to-pay (WTP) threshold of €20,000/QALY; this threshold was used, because it is recommended for prevention interventions by the Dutch National Healthcare Institute [37]. To address the cost-effectiveness at different WTP thresholds, the results of the Monte Carlo simulation were plotted in a cost-effectiveness acceptability curve. All plots were created using R and RStudio [25].

### 2.7. Budget Impact Analysis

To determine the short-term costs/savings of the implementation of this new dietary intervention, a BIA was performed in Microsoft Excel 365 (Microsoft Corporation, Redmond, WA, USA). The eligible population that can use this new intervention is all current and new RTR. The prevalence (*n* = 11,306) and incidence (904 new transplantations annually) were acquired from the Dutch Renal Registry (Renine) [38]. It was assumed that all patients will take the new dietary intervention, but that adherence will vary. A five-year time horizon was used with a payer’s perspective. To calculate the budget impact, the total number of patients was multiplied by the cost of the intervention per patient.

## 3. Results

### 3.1. Cost-Utility Analysis

To determine the cost and effects of implementing this new dietary intervention, a cost-effectiveness analysis was performed. The results for the adherence to the DASH diet and potassium supplementation are shown in Table 6 and Table 7, respectively. At a threshold of €20,000/QALY (the WTP threshold in the Netherlands for prevention interventions), both the DASH and potassium supplementation diets are cost-effective at 0% adherence to group 1. Potassium supplementation is dominant to the normal situation with incremental QALYs of 2901 and incremental costs of €−1,217,803. The DASH diet is dominant to the current standard of care with incremental QALYs of 1880 and incremental costs of €−27,934,786. With a stepwise increase in group 1 adherence of 10%, both dietary interventions remained cost-effective. To conclude, at a baseline of 0% adherence to group 1, both dietary interventions are dominant, and they remain cost-effective even if the adherence to group 1 increases.

The discounted costs for each of the different cost categories comparing the situation without the dietary intervention to the situation where there is 0% adherence to group, 1 for both the DASH diet and potassium supplementation, are shown in Appendix A, respectively. Most costs are caused by the upkeep of both the functioning graft and failed graft.

### 3.2. Probabilistic Sensitivity Analysis

To address possible uncertainty in the parameters used in the model, a PSA was performed. For the baseline adherence of 0% for group 1, a PSA was performed with 10,000 iterations for both the DASH and potassium supplementation diets. Figure 2A,B shows the cost-effectiveness planes for the DASH diet and for potassium supplementation, where it is shown that most outcomes fall either in the north-east or south-east quadrant. Cost-effectiveness acceptability curves have also been calculated for both simulations. With a WTP threshold of €20,000/QALY, 98.8% of outcomes are cost-effective for potassium supplementation, and 99.7% of outcomes are cost-effective for the DASH diet. With an adherence of 0% to group 1 and a WTP threshold of 0, 49.1% of outcomes are cost-effective for potassium supplementation, and 85.1% of outcomes are cost-effective for the DASH diet. The corresponding cost-effectiveness acceptability curves for an adherence of 0% to group 1 are also shown in Figure 2C,D To conclude, with a WTP threshold of €20,000/QALY, both the DASH and potassium supplementation diets are likely to be cost-effective or even cost-saving.

### 3.3. Deterministic Sensitivity Analysis

To determine whether single parameters had a strong effect on the ICER, a DSA was performed. Figure 3 shows the 15 most influential parameters in a tornado diagram. Values were compared to the baseline ICER, which was dominant for both the DASH diet and for potassium supplementation. For the DASH diet intervention, the upper limits of both the group 1 DWFG probability and the group 1 GF probability were the only parameters that caused a positive ICER due to the incremental costs becoming positive. The parameter with the strongest effect was the upper limit of the group 1 DWFG probability. All other parameters shown in Figure 3A had a limited effect on the ICER. For the potassium intervention, the effect on the ICER was lower when compared to the DASH. The parameters with the strongest effect were the group 1 DWFG and group 1 GF. All other parameters shown in Figure 3B had a limited effect on the ICER.

To conclude, the group 1 DWFG and group 1 GF probabilities had the strongest effect on the ICER in both the potassium and DASH interventions. However, for both the DASH diet and potassium supplementation, the ICER remained under the WTP threshold of €20,000/QALY for all parameters.

### 3.4. Budget Impact Analysis

To determine the total impact on the healthcare budget due to implementing this new dietary intervention, a BIA was performed that considered the total RTR population (*n* = 11,306) in the Netherlands with an annual inflow of 904. Costs were similar for the DASH and potassium interventions, because the dietary interventions remained the same. It was assumed that all current and new RTR would receive the intervention. The initial costs of implementing this intervention are €5,818,520, after which the annual costs will be €465,234. Over a five-year period, this amounts to a total of €8,144,693.

## 4. Discussion

The five-year cost of implementing dietary advice (either DASH or potassium supplementation) in the Dutch population are likely to stay below 10 million euros. Although the investment is quite extensive, savings might outweigh costs: our results show that adhering to the DASH diet and potassium supplementation increases the total amount of QALYs compared to the standard of care. Depending on the adherence, the intervention is either cost-saving or cost-effective. This suggests that from both an economic and healthcare point of view, introducing dietary advice into the Dutch RTR population could be a useful intervention.

To our knowledge, there are no studies looking at the cost-effectiveness of dietary interventions for RTR. There is a multicenter randomised controlled trial study currently in development, which aims to assess the effectiveness of exercise and diet on physical functioning, cardiometabolic health, and weight gain in RTR [39]. They will also gather data on cost-effectiveness. To address the patients’ diet, each patient in their intervention group will receive 15 months of dietary counselling with a total of 12 sessions. In our model, dietary counselling only took place in the first year with a total of eight sessions. It is unclear whether increasing the duration and number of visits will improve dietary adherence, and therefore the cost-effectiveness and budget impact, since there is no literature available on this subject. Furthermore, there are a limited number of studies looking at the cost-effectiveness of dietary interventions in reducing disease progression in CKD. One study found that a very low-protein diet is cost-effective when compared with a moderate low-protein diet [40]. Another study found that multidisciplinary care, which includes visiting a dietician, was cost-effective when compared to the normal situation for most patients with CKD [41]. Even though these models are different from the model used in this study, they still address the fact that it is cost-effective to delay dialysis.

With this cost-effectiveness analysis, we attempted to determine the total costs and effects of dietary interventions for different rates of adherence to group 1. At a baseline adherence of 0% to group 1 (and therefore the highest adherence rates to group 2 and 3), both the DASH diet and potassium supplementation were dominant; they increased the amount of QALYs and saved costs. This is in line with our expectations, because patients in groups 2 and 3 are less likely to lose a FG. With more patients keeping a FG, the total amount of QALYs is higher, and they have lower annual maintenance costs due to a kidney transplant upkeep being lower than the annual dialysis costs. However, in group 2 and group 3, patients are less likely to die with a FG, resulting in more patients staying alive; these patients require maintenance costs for a longer time, and it could be suspected that this could offset the cost saved due to fewer patients needing dialysis. Our results showed that even though more patients in group 2 and group 3 survived for a longer time, these costs were lower than the costs incurred due to dialysis. Another explanation is that patients who survive longer cause more costs in the future, but due to discounting these costs are relatively low. Contrary to this, the highest number of GFs occurs at the beginning of the model where discounting has a smaller effect.

Even though both treatments were dominant at baseline, we found that the DASH diet resulted in more savings, whereas potassium supplementation resulted in more QALYs. We had expected that an increase in QALYs would result in more savings, as can be seen with the DASH diet. This discrepancy is caused by differences in the transition probabilities of DWFG and GF between the diets. If we compare group 3 of both diets, we notice that the probability of GF is similar but that the probability of DWFG is higher for the DASH diet. This results in lower costs for the DASH diet and more QALYs for potassium supplementation, which seems to explain the fact that the DASH diet results in more savings and potassium supplementation in more QALYs.

It might be expected that the DASH diet and potassium supplementation will result in differences in adherence in practice [42,43,44]. An important reason for this could be related to the fact that it puts a financial burden on the patient as the costs for the DASH diet and potassium supplementation are currently not reimbursed by the Dutch authorities for RTR. This is a commonly seen problem in nutrition economics; the dietary intervention results in a financial burden on the patient whereas the savings mainly occur on the healthcare level. This complicates the interpretation of cost-effectiveness outcomes of dietary interventions. Moreover, although increasing the number of dietician visits from one to eight is expected to increase the adherence, the effect may be disappointing in practice. The DASH diet requires a different approach to cooking which might be challenging for an ageing population.

By dividing the group equally based on their adherence rates, we corrected for the fact that some of the patients do not achieve full adherence after the intervention, and therefore do not achieve maximal health effects. This was further explored by varying the percentage of patients that remained in group 1 after the intervention, that is, when fewer patients follow the diets correctly. Due to the aforementioned reasons, this is a more realistic view of how patients will adhere to the dietary intervention, since dietary adherence has been shown to be difficult to achieve and maintain [45]. The analyses showed that if the intervention has a positive effect on more than 10% and 20% of the patients in group 1 for the DASH diet and potassium supplementation respectively, the intervention would still be cost-saving.

However, this is based on the assumption that the adherence will remain equal over time. Especially the DASH diet is not easy to maintain [42]. Moreover, for both the dietary intervention and the potassium supplementation, it was assumed that there was an immediate effect of the intervention, while in reality it could take some time before an effect is achieved. We acknowledge this assumption might not entirely reflect reality and could potentially mitigate the cost-effectiveness results. For future clinical trials or observational cohort studies, it would be interesting to collect data on dietary adherence over time and to model this in a similar way as we did with the survival data.

Next to increasing the number of dietician visits, there are other ways to increase the adherence to diets. Monitoring of food intake has been shown to be a good predictor of dietary adherence, and several types of tools to assist in monitoring have been developed [46,47]. Several studies found that patients using a digital monitoring tool were significantly more likely to adhere to their dietary recommendations [48,49,50]. These methods have proven to be successful, but problems often arise due to uncertainty regarding meal size. Recently, an app (LogMeal) was developed based on deep learning and computer vision, which allows easy and objective monitoring of food intake [24]. The app allows for recognition of food in terms of type of dish, category, and ingredients. The volume of the dish can be estimated, and the micronutrient composition determined. Still, due to the novelty of LogMeal and the limited number of studies regarding long-term dietary adherence, it is unclear how well RTR will adhere to the diet. Our results showed that even with minimal improvements in dietary adherence the treatments remained cost-effective. This can be explained by the relatively low cost of the intervention quickly resulting in a cost-effective treatment.

The PSA showed that the results of our model are robust with 99.7% and 98.8% of simulations being cost-effective for the DASH diet and potassium supplementation, respectively. It is likely that the intervention will either be dominant to the current situation or cost-effective. The DSA showed that for both the DASH diet and potassium supplementation interventions, the parameters with the strongest effect were the probabilities of GF and DWFG, especially for group 1. The change in GF and DWFG probabilities directly affects the number of patients with a FG. Since this is the health state with the highest quality of life and the lowest cost, it is expected that differences in these probabilities will influence the ICER the most. Despite these effects, the interventions stayed either dominant or cost-effective for all parameters.

Evaluating the BIA, we found that the five-year costs of implementing the intervention were relatively low. The main component of the intervention is an increase in visits to the dietician: eight visits of around 30 min each or four hours in total. Dutch health insurance reimburses the first three treatment hours at the dietician for everyone [51]. This would mean that patients would only have to pay for the last two visits themselves, or they can extend their insurance to also cover this.

### 4.1. Limitations

To determine the costs of dialysis and transplantation, a study from 1998 was used [4]. It is likely that over time costs are going to change; however, the study by De Wit et al. is still the most elaborate published costing study regarding renal replacement therapy. It is still being used in recent studies regarding the cost-effectiveness of dialysis and kidney transplantation [7,23,52]. Moreover, based on the knowledge of a Dutch nephrologist, the costs used in our model are still correct.

To determine the transition probabilities in our model, patient level survival data was used. The data was extrapolated and several methods were used to calculate the annual transition probabilities for DWFG and GF [53]. Based on the advice of a nephrologist, at the end of our extrapolation the probability of dying should be higher. This can be addressed by using long-term survival data of RTR and combining this with the extrapolation data.

In our model, a decision was made against implementing the probability of receiving a pre-emptive re-transplantation. Normally, it is possible that when a patient already has a kidney transplant and this transplant starts failing, instead of first going to dialysis the patient pre-emptively receives a new kidney transplant. However, there is limited data regarding the probability of receiving a pre-emptive re-transplantation. For a normal pre-emptive transplantation in the Netherlands, the probability was 25% in 2017 [38]. However, it is reasonable to assume that this probability will drop sharply for the second transplant, with one possible reason being that relatives or friends are not able to donate a second kidney. If more data regarding the probability of a pre-emptive re-transplantation were to become available, it would be of interest to implement this in the model.

The number of events/outcomes in the model can be considered limited, which might affect the costs and outcomes of the model. However, we believe including more events would not have changed the general conclusion of the model, but would have increased both the complexity and uncertainty of the model. Since we focussed on the renal graft, for example cardiovascular events were not included in the model. Nevertheless, the DASH diet has shown to reduce the risk of cardiovascular events, and potassium supplementation could also potentially reduce these risks as well [54,55]. Therefore, the current outcomes of the model are likely a conservative estimate.

As this model uses retrospective data, the cost-effectiveness is mostly theoretical, since there are no trials comparing dietary adherence with an intervention and control arm.

### 4.2. Future Perspectives

The next step to be taken is to use RTR long-term survival data to improve the transition probabilities in the model. Doing so will make the model more robust and the results more credible. It is now of interest to look at whether we should focus more on the DASH diet or potassium supplementation. The DASH diet was dominant in all situations, as opposed to potassium supplementation, making it the preferred option. However, the adherence between the two diets could strongly differ: the DASH diet requires a change in diet whereas potassium supplementation only requires a pill. This difference could make the adherence of potassium supplementation much higher, thus making it the preferred option. It is also of interest to look at the combined effect of the DASH diet and potassium supplementation, because these two interventions could theoretically be combined into one dietary recommendation. This most likely would result in even more QALYs gained and costs saved. It is now necessary to investigate in a patient population how well patients will adhere to dietary interventions and to what degree their health outcomes will improve. This model can be used to justify those head-to-head studies, and it can be used to advocate for RTR receiving more visits to the dietician.

Moreover, as mentioned before, the long-term adherence to the DASH diet and potassium supplementation is uncertain. Therefore, a head-to-head trial could be used to investigate dietary adherence. Such trial could also possibly be used to support reimbursement of the DASH diet and potassium supplementation for RTR, so that the financial burden of the diet does not solely come down to the patient. It would be interesting to do a second cost-effectiveness analysis which could include adherence over time data.

## 5. Conclusions

To conclude, there is a limited number of available kidneys for transplantation. While patients wait for a kidney transplant, they often have no choice but to undergo dialysis to survive; this negatively affects their quality of life and is a burden on the healthcare budget. Once a patient receives a kidney transplant, it is important that it stays functional for as long as possible. We have shown that improvement of the adherence to the DASH and potassium supplementation diets in RTR is likely to be cost-saving and highly likely to be cost-effective and will improve quality of life by keeping the patients and the kidneys alive for longer. Therefore, the sooner patients improve their diet, the longer they and their kidneys will survive.

## Figures and Tables

**Figure 1 nutrients-13-01175-f001:**
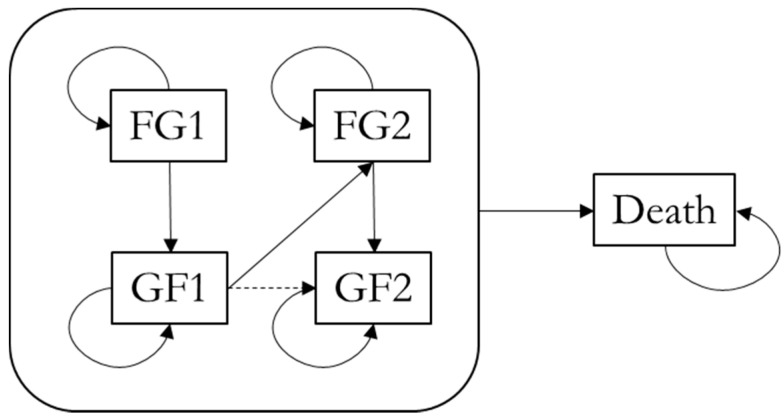
Markov model diagram. The dashed arrow indicates primary non-function. From each state, it is possible to transition to the Death state. Abbreviations: FG1, functioning graft 1; FG2, functioning graft 2; GF1, graft failure 1; GF2, graft failure 2.

**Figure 2 nutrients-13-01175-f002:**
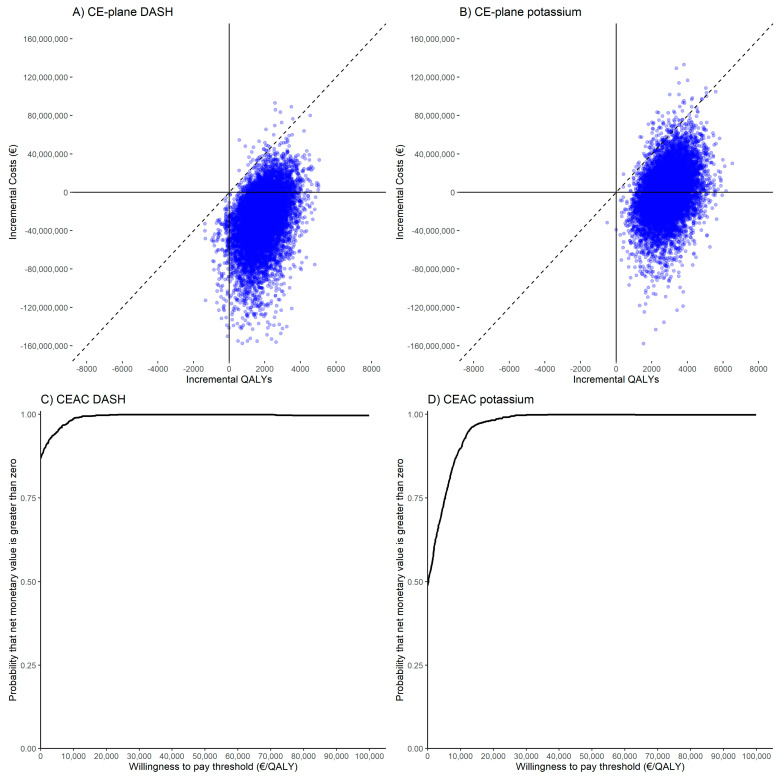
Probabilistic sensitivity analysis. (**A**) and (**B**) show the CE-plane of 10,000 Monte Carlo simulations with a willingness to pay threshold of €20,000/QALY. Uncertainty in the incremental total QALYs and costs is indicated by the dots. (**C**) and (**D**) represent the corresponding CEACs. Abbreviations: CE-plane, cost-effectiveness plane; CEAC, cost-effectiveness acceptability curve; DASH, Dietary Approach to Stop Hypertension; QALY, quality adjusted life-year.

**Figure 3 nutrients-13-01175-f003:**
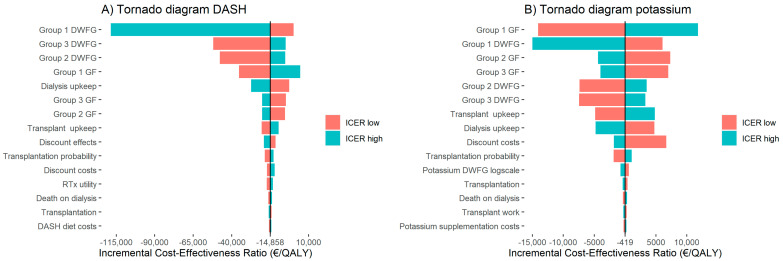
Deterministic sensitivity analysis. Panel (**A**) shows the tornado diagram for the DASH and panel (**B**) for potassium supplementation. The tornado diagram represents the effect of the lower and upper limit of the parameters on the ICER. The 15 most influential parameters are shown. Abbreviations: DASH, Dietary Approach to Stop Hypertension; DWFG, death with a functioning graft; GF, graft failure; QALY, quality adjusted life year; RTx, renal transplantation.

**Table 1 nutrients-13-01175-t001:** General parameters of the UMCG cohort.

Variable	Value (SD)
Number of patients	632
Age	53 (12.7)
Patient weight	80.3 (16.5)
Proportion male	56.3%
Proportion female	43.7%
Deceased donor kidney	65.7%
Living donor kidney	34.3%
Body Mass Index	26.65 (4.8)
Diabetes status	
Yes	24.2%
No	75.8%
Smoking Status	
Never	39.7%
Previously	44.1%
Current	12.2%
Unknown	4.0%
Number renal transplants	
1	89.6%
2	9.7%
3	0.6%
4	0.2%
Pre-emptive renal transplants	
Yes	83.7%
No	16.3%

Abbreviations: SD, standard deviation; UMCG, University Medical Centre Groningen.

**Table 2 nutrients-13-01175-t002:** Adherence to the DASH and to potassium intake without intervention and with intervention.

	Group 1	Group 2	Group 3
	DASH/Potassium	DASH/Potassium	DASH/Potassium
No intervention	100%	0%	0%
With intervention	0%	50%	50%
	10%	45%	45%
	20%	40%	40%
	30%	35%	35%
	40%	30%	30%
	50%	25%	25%
	60%	20%	20%
	70%	15%	15%
	80%	10%	10%
	90%	5%	5%

Abbreviation: DASH, Dietary Approach to Stop Hypertension.

**Table 3 nutrients-13-01175-t003:** Transition probabilities.

Variable	Value	Range	Source
Re-transplantation	0.150	0.077–0.224	Groen et al. [23]
Age of start decline	65	-	Assumption
Maximum age	80	-	Assumption
Primary non-function	0.007	0.004–0.011	UMCG dataset [18]
Death on dialysis			Pruthi et al. [24]
Age 50–54	0.040	0.035–0.046	
Age 55–59	0.052	0.045–0.058	
Age 60–64	0.076	0.069–0.083	
Age 65–69	0.102	0.093–0.110	
Age 70–74	0.138	0.127–0.150	
Age 75–79	0.190	0.176–0.205	
Age 80–84	0.240	0.219–0.262	
Age 85+	0.335	0.297–0.373	

Abbreviations: UMCG, University Medical Centre Groningen.

**Table 4 nutrients-13-01175-t004:** Costs and productivity losses.

Variable	Value	Range	Source
Intervention costs			
Number of dietician visits	8	2.7–13.2	Assumption
Cost per dietician visit	€60.00	€30.60–€89.40	Dutch healthcare insurance [31]
Travel costs per km	€0.19	-	Dutch National Healthcare Institute [27]
Average km to dietician	7	-	Dutch National Healthcare Institute [27]
Parking costs	€3	-	Dutch National Healthcare Institute [27]
Dietary adherence	0–100%	-	Assumption
Potassium supplementation costs			
Group 1	-	-	-
Group 2	€84.74	€38.68–€113.00	Dutch National Healthcare Institute [32]
Group 3	€169.47	€86.43–€252.51	Dutch National Healthcare Institute [32]
DASH costs			
Group 1	-	-	-
Group 2	€182.50	€93.08–271.93	Assumption, half of costs group 3
Group 3	€365	€186.15–€543.85	Expert opinion (€1 extra per day)
Costs per health state			
Dialysis upkeep	€87,855	€44,806–€130,904	De Wit et al. [4]
Transplantation upkeep	€10,820	€5518–€16,122	Groen et al. [23]
Transplantation	€59,949	€30,574–€89,324	De Wit et al. [4]
Death	€1165	€594–€1736	Groen et al. [23]
Renal graft failure	€2397	€1222–€3572	Groen et al. [23]
Discounting of costs	4%	0–8%	Dutch National Healthcare Institute [27]
Productivity losses			
Percentage of patients working in the functioning graft state			
Age 45–54	57%	29–85%	Jansen et al. [33]
Age 55–64	41%	21–61%	Jansen et al. [33]
Age 65+	6%	3–9%	Assumption (Same decline as in general population)
Percentage of patients working in the graft failure state			
Age 45–54	32%	16–48%	Jansen et al. [33]
Age 55–64	19%	10–28%	Jansen et al. [33]
Age 65+	3%	1–4%	Assumption (Same decline as in general population)
Days in hospital	10.58	5.40–15.80	UMCG year report [34]
Friction period	85	-	Dutch National Healthcare Institute [27]
Workhours per day	8	-	Dutch National Healthcare Institute [27]
Male hourly salary	€37.90	-	Dutch National Healthcare Institute [27]
Female hourly salary	€31.60	-	Dutch National Healthcare Institute [27]

Abbreviations: UMCG, University Medical Centre Groningen.

**Table 5 nutrients-13-01175-t005:** Utility values.

State	Value	Range	Source
HD	0.56	0.49–0.62	Liem et al. [36]
PD	0.58	0.5–0.67	Liem et al. [36]
Functioning graft	0.81	0.72–0.9	Liem et al. [36]
Death	0	0	-
Discounting of effects	1.5%	0–3%	Dutch National Health Care Institute [27]

Abbreviations: HD, haemodialysis; PD, peritoneal dialysis.

**Table 6 nutrients-13-01175-t006:** Adherence to the different DASH groups and the respective incremental costs, effects, and ICER.

Group 1	Group 2	Group 3	Incremental Costs	Incremental Effects (QALY)	ICER (€/QALY)
100%	0%	0%	-	-	-
0%	50%	50%	€−27,934,786	1880	Dominant
10%	45%	45%	€−25,089,794	1692	Dominant
20%	40%	40%	€−22,244,801	1504	Dominant
30%	35%	35%	€−19,399,808	1316	Dominant
40%	30%	30%	€−16,554,816	1128	Dominant
50%	25%	25%	€−13,709,823	940	Dominant
60%	20%	20%	€−10,864,830	752	Dominant
70%	15%	15%	€−8,019,838	564	Dominant
80%	10%	10%	€−5,174,845	376	Dominant
90%	5%	5%	€−2,329,853	188	Dominant

Abbreviations: DASH, Dietary Approach to Stop Hypertension; ICER, Incremental Cost-Effectiveness Ratio; QALY, quality-adjusted life year.

**Table 7 nutrients-13-01175-t007:** Adherence to the different potassium groups and the respective incremental costs, effects, and ICER.

Group 1	Group 2	Group 3	Incremental Costs	Incremental Effects (QALY)	ICER (€/QALY)
100%	0%	0%	-	-	-
0%	50%	50%	€−1,217,803	2901	Dominant
10%	45%	45%	€−1,044,508	2611	Dominant
20%	40%	40%	€−871,214	2321	Dominant
30%	35%	35%	€−697,920	2031	Dominant
40%	30%	30%	€−524,626	1741	Dominant
50%	25%	25%	€−351,331	1451	Dominant
60%	20%	20%	€−178,037	1160	Dominant
70%	15%	15%	€−4,743	870	Dominant
80%	10%	10%	€168,552	580	€290
90%	5%	5%	€341,846	290	€1178

Abbreviations: ICER, Incremental Cost-Effectiveness Ratio; QALY, quality-adjusted life year.

## Data Availability

The data presented in this study are available on request from the corresponding author. The data are not publicly available due to ethical requirements.

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
