# Peer review of "Modelling the Cost-Effectiveness of Implementing a Dietary Intervention in Renal Transplant Recipients"

_nutrients, 2021, doi:10.3390/nu13041175_

Round 1

Reviewer 1 Report

Thanks for your response. I've no further comments.

Author Response

Dear reviewer, 

Please find our response to your comments in the attached document. 

Thank you again for taking the time to review our manuscript. 

Kind regards,

Lisa de Jong

Reviewer 2 Report

Transition probabilities:

it is still not clear to me how and where the DASH or potassium supplementation effect on the transitions have been collected. It says patient data and literature. So moving from insufficient to recommended K intake would "shift" the transition. Where are these values? It furthermore looks like it is "immediate". It could be as K turnover is rather high but this has to be clearly stated. 

Future perspective: 

Adequate K intake is not only good for renal health but even more important for cardiovascular events. Especially in this cohort which looks like high risk (high diabetes, overweight, older). I understand this study focuses on renal transplant but because cardiovascular is a competing event it should at least have been mentioned. 

on the fact that patients "pay" the nutritional preventive intervention

this is quite a central point. The authors use a societal approach but the incremental costs of DASH/supplementation is on the patients. The opportunity costs (of cooking) and level of adherence is also at the patient level. It would have been important to calculate the CUE by groups (patients, healthcare etc). This problem is actually the core issue with nutrition economics. Patients pay for the nutrition interventions and healthcare "gets" the health effects (well of course is the QALY for patients). Furthermore because it is a rather small risk reduction, the positive effect of nutritional "prevention" is difficult to maintain for patients. Especially in this age group. Adherance of nutritional changes is very tricky. Despite 20y of public health intervention in the Netherlands, salt intake is still too high, especially in this age group. It would have been nice to use the "learning" from the salt campaigns to estimate how this adherence to an increased K intake would mean. We know that the NaCl decreases in the UK and NL have not come from consumers changes but from product reformulation especially in groups of population with low nutritional awareness (men, older population etc). The assumption that this group will be compliant to change radically its nutrition (DASH is really not easy to maintain) is very optimistic. Here the authors should have also done some literature research on "adherence over time to the DASH diet).  I do however feel that indeed a pill would probably be easier to do (especially as this population is probably already taking some form of medication). 

Author Response

(The authors gave the same response as above.)

Reviewer 3 Report

The authors revised it according to our suggestions. We have no claim in the revised paper.

Author Response

Dear reviewer, 

Please find our response to your comments in the attached document. 

Thank you again for taking the time to review our manuscript. 

Kind regards,

Lisa de Jong

This manuscript is a resubmission of an earlier submission. The following is a list of the peer review reports and author responses from that submission.

Round 1

Reviewer 1 Report

The authors reported effect of DASH and potassium supplementation for the risk of death with a functioning graft and renal graft failure in RTR.

The authors should show the characteristics of all patients. For example, body mass index, diabetes, hypertension, hyperlipidemia, smoking status..... 

Author Response

Dear editor, 

Please find our response to reviewer 1 in the attached word file. 

Kind regards,

Lisa de Jong

Reviewer 2 Report

My main remark is that the authors -as they use a societal approach- need to take the costs of the DASH diet and the supplementation costs of potassium. These costs are significant. If i use Monsivais et al 2015 (https://academic.oup.com/ajcn/article/102/1/138/4564238) the incremental costs of the DASH diet are 20%. Over a life time this would be significant. Furthermore the main reason of the low compliance of the DASH diet is that it is also a significant change in taste and cooking time. I am not sure how successful a diet change in an aging population would be? And who would cook all the additional vegetables etc? Yes older citizens have more time to cook but do they really (especially men)? At least it should be mentioned in the discussion. 

Costs of a potassium supplementation also has be included. A quick search gave me yearly costs of 1000 Dollars. This is probably too high but it needs to be included. What would also need to be done is "diagnostics" and "tracking". Additional potassium for these patients could be a risk meaning that the level of supplementation needs to be closely monitored. 

I wonder if a cohort markov model would not have been more appropriate as there seems be be significant heterogeneity in the cohort (age, pre existing conditions). Because the authors have this large cohort these more granular transitions would be possible. 

What is not really clear is how deficient the individuals are in terms of potassium. Potassium intake is not that bad in Northern Europe. Would an increase of K intake really decrease the risks?

Please write chronic kidney disease at least once. 

 I understand that in the NL societal risks are the preferred approach but it would have been interesting to split the costs and savings by payers, patients and "labour market". Who would pay the costs of the DASH diet/K supplementation?

Author Response

Dear editor, 

Please find our response to reviewer 2 in the attached word file. 

Kind regards,

Lisa de Jong

Reviewer 3 Report

This is an excellent manuscript in the field of health economics. Many assumptions support the results.

1. However, we cannot apprehend the impact of DASH on renal outcomes and how to improve the adherence of diet control clinically. It focus on cost-effectiveness theoretically but without intervention and control groups in practice.

2. It is likely to increase the adherence through increasing the number of visits to the dietician from one to eight dietician visits. But it always disappointed us really, the adherence of patients may corelated poorly the numbers of visits.

3. There is no other comparable factors or events discussed, hospitalization from infection early after transplantation for example; will influence the renal and patient outcomes for those renal transplant patients. 

Author Response

Dear editor, 

Please find our response to reviewer 3 in the attached word file. 

Kind regards,

Lisa de Jong
